# The Cross Marks the Spot: The Emerging Role of JmjC Domain-Containing Proteins in Myeloid Malignancies

**DOI:** 10.3390/biom11121911

**Published:** 2021-12-20

**Authors:** Hans Felix Staehle, Heike Luise Pahl, Jonas Samuel Jutzi

**Affiliations:** 1Division of Molecular Hematology, University Medical Center Freiburg, Faculty of Medicine, University of Freiburg, 79098 Freiburg, Germany; hans.felix.staehle@uniklinik-freiburg.de (H.F.S.); heike.pahl@uniklinik-freiburg.de (H.L.P.); 2Division of Hematology, Department of Medicine, Brigham and Women’s Hospital, Harvard Medical School, Boston 02115, MA, USA

**Keywords:** KDM, Jumonji C (JmjC) domain, histone demethylation, leukemia, myelodysplastic syndrome, myeloproliferative neoplasm, targeted therapy

## Abstract

Histone methylation tightly regulates chromatin accessibility, transcription, proliferation, and cell differentiation, and its perturbation contributes to oncogenic reprogramming of cells. In particular, many myeloid malignancies show evidence of epigenetic dysregulation. Jumonji C (JmjC) domain-containing proteins comprise a large and diverse group of histone demethylases (KDMs), which remove methyl groups from lysines in histone tails and other proteins. Cumulating evidence suggests an emerging role for these demethylases in myeloid malignancies, rendering them attractive targets for drug interventions. In this review, we summarize the known functions of Jumonji C (JmjC) domain-containing proteins in myeloid malignancies. We highlight challenges in understanding the context-dependent mechanisms of these proteins and explore potential future pharmacological targeting.

## 1. Introduction

Non-genetic chromatin alterations play a significant role in disease development, maintenance, and relapse of myeloid malignancies. Among epigenetic modifying enzymes, histone demethylases have gained special attention in recent years for their involvement in myeloid malignancies. Histone lysine demethylases (KDMs) are subdivided into two subclasses, FAD-dependent and Jumonji C (JmjC) domain-containing (JMJD) demethylases (Figure 1). Although there are only two FAD-dependent demethylases, KDM1A and KDM1B, JMJD demethylases comprise a larger group of more than twenty proteins [1,2,3]. Based on the evolution of their JmjC domain and the occurrence of additional domains, JMJD demethylases are subdivided into numbered subgroups (KDM2–7). Most KDMs are attributed to demethylate methylated lysines of histone 3 or 4, as well as of non-histone proteins, thereby activating or repressing transcription. KDM1A has been studied extensively, culminating in the development of inhibitors currently in clinical trials for patients with leukemia [2]. The JmjC domain containing KDMs, which require Fe(II) and a-ketoglutarate (a-KG) as cofactors, however, have been less intensely studied and their substrate specificity varies greatly [3,4,5]. Our lack of knowledge regarding the function of these KDMs in benign hematopoiesis makes understanding their role in malignancies even harder. Moreover, the exact contribution of individual KDMs in either a tumor-suppressive or tumor-promoting manner, is dependent on the specific disease-context. In this review, we will summarize current knowledge of KDM2–7 protein function both in normal hematopoietic stem cells and in myeloid malignancies. We highlight the potential use of KDM expression levels as biomarkers and discuss the implications for future therapeutic targeting of KDMs in myeloid malignancies.

## 2. Results

For all abbreviations of important proteins used in the context of normal hematopoiesis and myeloid malignancies, please refer to Appendix A.

### 2.1. The KDM2 Family

The KDM2 subfamily of the Jumonji domain-containing histone demethylases encompasses two members, KDM2A and KDM2B. As reviewed by Markolovic et al., both enzymes possess catalytic activity towards mono- and di-methylated lysine residues [8]. H3K36 is the known main histone target of KDM2 demethylases [8]. In myeloid malignancies both enzymes have been shown to demethylate H3K36me^2^ [10,11,12].

#### 2.1.1. KDM2A

Cumulating evidence indicates a tumor-suppressor role for this histone demethylase in myeloid leukemias (Figure 2) [10,13,14]. KDM2A is required for the maintenance of heterochromatin, as its deletion delocalizes HP1 from chromatin, impairs centromeric integrity and, thereby, increases genomic instability [14]. Fittingly, Inoue et al. observed KDM2A downregulation in the bone marrow (BM) of mice with benzene-induced acute myelogenous leukemia (AML) compared to control BM (Table 1) [13]. Functional evidence provided by Zhu et al. showed that KDM2A reduces expression of Hoxa9 and other MLL target genes in MLL-AF10 mice, resulting in increased differentiation [10]. MLL is a H3K4 histone methyltransferase and MLL gene rearrangements are common chromosomal abnormalities associated with acute leukemias [15]. Mechanistically, KDM2A reversed H3K36me^2^ methylation, a mark written by the histone methyltransferase ASH1L to facilitate binding of MLL and its associated oncoprotein LEDGF (Table 1) [10]. KDM2A activity thus counteracts MLL-driven leukemogenesis, underlining an anti-leukemic role for KDM2A in myeloid leukemias.

#### 2.1.2. KDM2B

Two complementary studies suggested an oncogenic role for KDM2B in myeloid leukemia (Figure 2) [11,12]. KDM2B demethylates H3K36me^2^ at the promoter of the tumor suppressor p15 (Ink4b), leading to decreased expression. p15 (lnk4b) silencing leads to cell-cycle progression in both AML cell lines and in CD34^+^ primary AML cells [11] and supports AML transformation by Hoxa9/Meis1 in mice (Table 1) [12]. Moreover, transgenic (tg) mice overexpressing KDM2B develop myeloid and B cell leukemias through a KDM2B-mediated increase in Nsg2 and OXPHOS expression [17]. These mice show impaired HSC differentiation and dysregulation of metabolic processes. Nsg2 impairs HSC differentiation, while upregulation of OXPHOS genes provides a metabolic proliferative advantage for tg Kdm2b hematopoietic stem cells (HSCs) [17]. As a member of the non-canonical polycomb repressive complex 1.1 (PRC1.1) KDM2B also contributes to metabolic dysregulation. PRC1.1 controls the expression of genes essential for leukemogenesis, for example pyruvate kinase and lactate dehydrogenase, and is required for the viability of MLL-AF9 driven murine leukemias, as well as of primary patient AML cells (Table 1) [16]. These data thus suggest an oncogenic role for KDM2B.

Contradictory findings, however, support a tumor-suppressor role for KDM2B in myeloid malignancies (Figure 2). Karoopongse et al. showed that low expression of KDM2B, seen in primary MDS cells, allows upregulation of let-7b, a microRNA that targets EZH2 mRNA for degradation. Vice versa, KDM2B overexpression rescues EZH2 expression in MDS cell lines by suppressing let-7b [18]. EZH2 functions as a tumor suppressor [66,67], clinically apparent by the poor prognosis of patients with AML/MDS with reduced EZH2 activity [68,69]. Andricovich et al. likewise described an association between low expression of KDM2B and reduced survival of AML patients [19,70,71]. Moreover, this finding was confirmed in a Vav1-Cre Kras^G12D^-driven AML mouse model. Mice displayed shortened survival upon loss of Kdm2b, while overexpression significantly extended survival of Kras^G12D^-induced AML [19]. Mechanistically, loss of Kdm2b resulted in cell-cycle activation and reduced interferon signaling. Overexpression led to the activation of transcriptional programs linked to interferon signaling (Irf3, Irf5, Irf7, Stat1, Stat2) and inhibition of the PRC2 antagonists Hox10 and Smarca4/Brg1 (Table 1) [19]. In myeloid disorders, the PRC2 complex is associated with tumor suppressive activity [72,73]. In several model systems KDM2B acts as a tumor suppressor, by preventing degradation of a second tumor-suppressor, by inhibiting tumor-suppressor antagonists, or by reducing cell activation. In conclusion, the role of KDM2B in myeloid malignancies appears to be strongly context dependent. Future studies exploring any potential therapeutic targeting of KDM2B will need to consider that KDM2B either acts as an oncogene or a tumor suppressor depending on the specific genetic background.

### 2.2. The KDM3 Family

The KDM3 subfamily of the Jumonji domain-containing histone demethylases encompasses three members, KDM3A, KDM3B, and KDM3C. As reviewed by Markolovic et al., all members possess catalytic activity towards mono- and di-methylated lysine residues [8]. H3K9 is the known main histone target of KDM3 demethylases [8]. In myeloid malignancies, KDM3A has been shown to demethylate H3K9me^2^ [20], while KDM3B is associated with histone demethylase activity towards H3K9me^1–2^ [21,22,30]. The catalytic function of KDM3C in myeloid malignancies is controversial. Histone demethylase activity towards H3K9me^1–3^ [28,32] and H3K36me^3^ [28] has been observed, while other groups propose a non-enzymatic role in this context [29,31].

#### 2.2.1. KDM3A

Jafek et al., demonstrate that KDM3A supports pro-leukemic processes in AML (Figure 2). Oct1 recruits KDM3A to the CDX2 promoter to remove the repressive H3K9me2 mark [20]. Overexpression of CDX2 is often observed in AML patients and is sufficient to induce leukemia in murine BM transplantation models [74,75]. To date, this is the only study describing a role for KDM3A in myeloid malignancies; further investigation is therefore required to corroborate the significance of KDM3A-mediated CDX2 overexpression in this context or to reveal additional pathophysiological roles for KDM3A.

#### 2.2.2. KDM3B

KDM3B has been proposed to act as a tumor suppressor in myeloid disorders (Figure 2). Located on chromosome 5, an allele of KDM3B is lost upon 5q deletion, frequently observed in AML and MDS [23,76]. In AML cell lines harboring the 5q deletion, reintroduction of KDM3B represses clonogenic growth and colony formation) [22,23]. Moreover, in APL cell lines, KDM3B downregulation enhanced proliferation, impaired differentiation, and reduced ATRA-induced degradation of PML/RARα [21]. Mechanistically, KDM3B modulates H3K9me^1–2^ levels to maintain a compact chromatin status, thus, loss of KDM3B enhances chromatin accessibility [21,22]. Open chromatin allows PML/RARa access to ETS and bZIP transcription factor binding sites, among them the sites of SPI1/PU.1 and Jun-AP1 [21]. The SPI1/PU.1-mediated gene signature is essential for myeloid differentiation, and this is repressed upon PML/RARA binding [77,78]. Therapy with ATRA degrades PML/RARA. In APL cell lines, this process is accompanied by KDM3B upregulation, which enhances gene expression required for differentiation through its demethylase activity [21,22]. Taken together, KDM3B exerts anti-leukemic effects in AML, especially APL, and MDS by modulating chromatin accessibility through its enzymatic activity.

#### 2.2.3. KDM3C

Research of the last decade revealed an oncogenic role for KDM3C in myeloid leukemias (Figure 2). Using an shRNA-mediated depletion screen Sroczynska et al. identified Kdm3c as one of the best candidate drug targets for leukemia therapy among 319 tested genes. In AML cell lines and in an MLL-AF9 mouse model, depletion of Kdm3c impaired growth and colony formation of leukemic cells by increasing apoptosis [33]. Further studies in the following years confirmed the finding by Sroczynska et al. and revealed catalytic and non-catalytic mechanisms through which KDM3C exerts its oncogenic potential.

Several groups have proposed that catalytic function of KDM3C is required for its pro-leukemic activity. In RUNX1-RUNX1T1-driven leukemia, KDM3C is directly recruited to its target genes by RUNX1-RUNX1T1. KDM3C-mediated demethylation of H3K9me^2^ at the promoters results in increased expression of its targets among them LMO2, ID1, EGR1, and CDKN1A. Vice versa, shRNA-mediated depletion of KDM3C reduced target gene expression and impaired proliferation and survival in AML cell lines [32]. Likewise, Izaguirre-Carbonell et al. showed that the catalytic JmjC-domain and the zinc finger domain of KDM3C are required for the survival of MLL-AF9 leukemia cells. Silencing of KDM3C promoted cell differentiation through upregulation of the IL3 receptor, followed by enhanced RAS/MAPK and JAK-STAT signaling) (Table 1) [28]. In this model, loss of KDM3C activity increased H3K36me^3^ levels while H3K9^1–3^ methylation remained unaltered [28]. Despite reports showing non-enzymatic modes of action for KDM3C, these findings clearly show that KDM3C sustains leukemogenicity of AML cells through its enzymatic activity.

Other groups have suggested that the pro-leukemic role of KDM3C is independent of its histone demethylase activity. Zhu et al. showed that a knockout of KDM3C in MLL-AF9 and HOXA9-driven leukemic mice increased overall survival by reducing the number of leukemia initiating cells (LSCs) and promoting differentiation of leukemic cells (Table 1) [31]. Mechanistically, KDM3C modulates a HOXA9-controlled gene-expression program that is independent of histone methylation [31]. Instead, KDM3C directly interacts with HOXA9 and may serves as a scaffold to facilitate binding between HOXA9 and its cofactors [31]. Additional evidence for a demethylase-independent role in AML derives from overexpression of an enzymatically compromised isoform of KDM3C, lacking the zinc finger domain. Enzymatically inactive KDM3C intensifies the aggressive phenotype of HOXA9 leukemia in mice similar to the wt protein. Neither wildtype (wt) KDM3C nor the defective KDM3C isoform showed demethylase activity towards H3K9 [29]. Nonetheless, both upregulated expression of key glycolytic and oxidative enzymes. KDM3C therefore also contributes to AML pathophysiology independent of its enzymatic activity.

The oncogenic role of KDM3C was also tested in myeloproliferative neoplasms (MPN). In Ph-positive CML cell lines, depletion of KDM3C impairs proliferation and viability and enhances sensitivity towards chemotherapy treatment [25]. In Ph-negative MPN, KDM3C is regulated by NFE2, a key transcription factor in aberrant MPN signaling [30,79,80,81,82,83]. KDM3C increases the expression of NFE2 by reducing H3K9me^1/2^ at the NFE2 promoter region, resulting in a positive feedback loop [30]. Knockdown of Kdm3c in a hematopoietic cell line carrying Jak2^V617F^ significantly reduces cytokine-independent proliferation. However, Jak2^V617F^-driven MPN disease initiation was not altered in Kdm3c knockout mice [26,30]. Since there are controversial findings towards the role of KDM3C in MPN, further investigation is required to clarify the role of KDM3C and its potential as a pharmacological target in MPN.

An increasing number of KDM3C inhibitors are entering pre-clinical investigation. The Hu laboratory performed a virtual screen for potential small molecular modulators targeting the Jumonji domain of KDM3C [24,27]. JDI-16 exhibited killing activities against malignant hematopoietic cells and induced apoptosis and differentiation of MLL rearranged AML cells [27]. In addition, the KDM3C modulator JDM-7 suppressed colony formation of AML cell lines in semi-solid cell culture by decreasing HOXA9-mediated expression patterns [24]. Pre-clinical data thus provide efficacy of KDM3C inhibition in mouse models of AML. Future studies are necessary to determine clinical applicability of KDM3C inhibitors.

### 2.3. The KDM4 Family

The KDM4 subfamily of the Jumonji domain-containing histone demethylases encompasses five members, KDM4A, KDM4B, KDM4C, KDM4D, and KDM4E. As reviewed by Markolovic et al., all enzymes possess catalytic activity towards mono-, di-, and tri-methylated lysine residues [8]. H3K9, H3K36, and H1.4K26 are the known main histone target of KDM4 demethylases [8]. In myeloid malignancies H3K9me^3^ demethylase activity was described for KDM4A-D [34,35,39,40,41,42]. For KDM4A and KDM4C additional activity was observed towards H3K27me^3^ [35,40]. The role of KDM4E in myeloid malignancies remains unknown.

Cumulative evidence from the Helin lab supports the hypothesis that KDM4 family members exert redundant demethylase activities, suggesting compensation for loss of single KDM4 proteins. The Helin lab showed that only a combined knockout of several histone demethylases, especially the combination of Kdm4a and Kdm4c, impaired the maintenance of HSCs [84,85,86]. Furthermore, only simultaneous knockout of the demethylases Kdm4a, Kdm4b, and Kdm4c perturbed progression of MLL-AF9 translocated leukemias in mice [34]. Mechanistically, KDM4 proteins remove repressive H3K9me^3^ to drive their target gene expression, including Taf1b, Nom1, and Il3ra [34,84,85,86]. These genes are essential for normal hematopoiesis and are often deregulated in AML [87,88,89]. Disease-modification in myeloid malignancies through targeting of KDM4 demethylases thus appears feasible but will require the simultaneous inhibition of several enzymes.

Other groups also suggest a pro-leukemic role for KDM4 demethylases (Figure 2). However, in contrast to the Helin laboratory, these researchers do not see the requirement for combined targeting to impair leukemogenesis:

#### 2.3.1. KDM4A

Massett et al. show that KDM4A is required for the survival of AML cell lines and AML primary cells (Table 1) [35]. Loss of KDM4A alone is sufficient to induce apoptosis and transplantation of human KDM4A-depleted MLL-AF9 AML cells into immunodeficient mice significantly prolongs survival compared to the control group [35]. Mechanistically, KDM4A enhances the expression of PAF1 by reducing H3K9me^3^ and H3K27me^3^ at its promoter region. PAF1 is essential for transcriptional elongation as a core component of the polymerase-associated factor 1 (PAF1) complex [36,90,91]. PAF1 and KDM4A initiate a pro-leukemic transcriptional program that involves upregulation of pro-survival BCL2 and downregulation of pro-apoptotic BCL2L11 (BIM) [35]. Detailed analysis of this transcriptional program revealed a gene signature that effectively stratifies high-risk AML patients [35]. In addition, newly developed KDM4A inhibitors efficiently block enzymatic activity and show antitumor activity in solid and hematological cancer cell lines [92]. Thus, KDM4A is a promising target for novel diagnostic and therapeutic approaches in myeloid malignancies.

#### 2.3.2. KDM4B

KDM4B expression is increased in both AML patients and in MLL-AF9-transduced cord blood CD34^+^ cells, indicating an involvement of KDM4B in AML leukemogenesis (Figure 2) [37]. Mechanistically, MLL-AF9 enhances the expression of S100A8 and S100A9 which inhibit differentiation in AML [93]. shRNA-mediated knockdown of KDM4B in THP-1 and MLL-AF9 transduced CD34^+^ cells significantly reduced growth of leukemic cells and normalized S100A8/9 expression [37]. These data support the hypothesis that MLL-AF9 leads to activation of S100A8/9 through KDM4B. However, whether this occurs through direct demethylase activity at the S100A8/9 loci or by an indirect mechanism is unclear.

#### 2.3.3. KDM4C

KDM4C is upregulated in AML primary cells and in AML cell lines (Table 1) [38,39]. Moreover, this histone demethylase is involved in various processes essential for AML pathophysiology. Xue et al. showed that KDM4C contributed to cytarabine resistance in AML via regulation of the MALAT1/miR-328-3p/CCND2 axis [38]. In resistant HL-60 AML cells (HL-60/A), KDM4C elevates the expression of MALAT1, a long non-coding RNA that is associated with increased cell proliferation in AML [94]. MALAT1 allows for the increased CCND2 expression through inhibition of the microRNA miR-328-3p [38]. As a member of the cyclin family CCND2 has been shown to confer drug resistance in cancer [95]. Notably, transplantation of KDM4C depleted HL-60/A cells into immunodeficient NOD/SCID mice significantly improved survival compared to KDM4C-expressing cells [38]. These results underline the potential for novel therapeutic approaches targeting KDM4C in cytarabine-resistant AML.

Wang et al. described KDM4C as part of the KDM4C/ALKBH5/AXL axis, providing additional evidence of its contribution to AML. In AML cell lines, KDM4C allows for the increased expression of the N^6^-methyladenosine (m^6^A) demethylase ALKBH5 through removal of repressive H3K9me^3^ at its promoter [39]. ALKBH5 increases mRNA stability of the receptor tyrosine kinase AXL, resulting in enhanced PI3K/AKT/mTOR signaling [39]. Fittingly, knockdown of KDM4C in MOLM13 and THP1 cells inhibited cell growth and clonogenic ability through the downregulation of ALKBH5 [39]. Thus, KDM4C contributes to aberrant signaling in AML.

Moreover, KDM4C is involved in the leukemic network mediated by PRL-3 [41], an oncogenic dual-specificity phosphatase frequently overexpressed in AML patients [96]. In TF-1 cells, PRL-3 promotes KDM4C occupancy of the Leo1 promoter, which is accompanied by a reduction in H3K9me^3^ repressive marks and increased Leo1 expression [41]. Leo1 is part of the RNA polymerase II–associated factor (PAF) complex and mediates oncogenic activities in AML [41]. Hence, KDM4C regulates the expression of the oncogenic PRL-3 target gene Leo1 via its histone demethylase activity.

Additionally, Cheung et al. show that KDM4C is essential for MLL- and MOZ-TIF2-driven leukemia. shRNA-mediated knockdown of KDM4C in MLL-GAS7, MLL-AF9, and MOZ-TIF2 leukemia cell lines increased differentiation, cell cycle arrest, and apoptosis [40]. Moreover, transplantation of KDM4C-depleted leukemic cells into sub-lethally irradiated mice significantly improved their survival [40]. Mechanistically, MLL-fusions and MOZ-TIF2 recruit KDM4C to remove repressive H3K9me^3^ and H3K27me^3^ histone marks at their target genes, among them Myc, Hoxa9, and Meis1. [40]. Consistently, loss of Kdm4c was accompanied by reduction in H3K9 acetylation, as well as accumulation of H3K9me^3^ and H3K27me^3^ [40]. Furthermore, treatment with the KDM4C inhibitor SD70 [97] drastically reduced the leukemic burden and significantly extended disease latency in mice transplanted with MLL-AF9 leukemia cells [40]. Multiple findings demonstrate a key role for KDM4C in MLL- and MOZ-TIF2-driven leukemia. This demethylase therefore represents a promising candidate in this challenging subtype of myeloid leukemia and should be further investigated in preclinical and clinical studies to test its applicability as potential drug target.

#### 2.3.4. KDM4D

Wu et al. recently suggested a role for KDM4D in AML pathophysiology (Table 1, Figure 2). Using the databases SurvExpress and GEPIA, they found high expression of KDM4D to be significantly correlated with a poor prognosis in AML [42]. Fittingly, KDM4D is highly expressed in HL-60, MOLM-13, and NB4 AML cells. Overexpression or downregulation of KDM4D in these cell lines caused an increase or decrease in proliferation, respectively [42]. Mechanistically, KDM4D increases the expression of the anti-apoptotic MCL1 through removal of repressive H3K9me^3^ [42]. This study is the first to propose involvement of KDM4D in AML.

### 2.4. The KDM5 Family

The KDM5 subfamily of the Jumonji domain-containing histone demethylases encompasses four members, KDM5A, KDM5B, KDM5C, and KDM5D. As reviewed by Markolovic et al., all enzymes possess catalytic activity towards mono-, di-, and tri-methylated lysine residues [8]. H3K4 is the known main histone target of KDM5 demethylases [8]. In myeloid malignancies KDM5A and KDM5B show enzymatic activity towards H3K4me^3^ [47,98] while little is known about KDM5C [51,52] and KDM5D in this context.

#### 2.4.1. KDM5A

The function of KDM5A was recently comprehensively reviewed by Kirtana et al. [99]. The role of KDM5A in myeloid malignancies is best understood in the context of NUP98 fusions. NUP98 is frequently fused to a variety of genes in pediatric acute leukemias through translocations or inversions, for instance it can fuse to the PHD domain of KDM5A [43,100], forming NUP98-KDM5A. These fusions increase expression of the Hoxa gene cluster by binding of promoters. The KDM5A PHD domain exerts a histone reader function, which is redirected to NUP98 targets through the aberrant fusion. As a result, regular myelo-erythroid differentiation is perturbed [101,102,103,104,105,106,107,108]. Consequently, transplantation of BM infected with a NUP98-KDM5A fusion induced the development of AML in mice (Table 1) [43].

The mode of action of KDM5A fusions is distinct from the enzymatic function of the physiological JmjC domain in full-length KDM5A, which acts as a histone eraser. It is therefore also important to understand how deregulated expression of full-length KDM5A contributes to disease evolution and progression. In solid tumors KDM5A is overexpressed and contributes to tumorigenesis by reduction in differentiation, to metastasis, and to drug resistance [98,101,109,110,111,112,113]. However, little is known about the role of full-length, enzymatically active, KDM5A in myeloid malignancies. In chronic myelogenous leukemia (CML) blast phase (CML-BP), KDM5A plays an important role [44,45], bar one exception. MicroRNA-181d (miR-181d) is overexpressed in CML-BP, which directly targets and deregulates the expression of KMD5A. Elevated KDM5A levels in turn inhibit the transcriptional repression of NF-kB subunit p65, a direct KDM5A target. The ensuing increase in NF-kB activation is further augmented by a positive feedback loop through binding of p65 to the miR-181d promoter [45]. This sustained activation suggests both miR-181d and KDM5A as potential therapeutic targets in CML-BP.

Another study described KDM5A downregulation in CML-BP and found that KDM5A downregulated miR-21 expression by demethylation of H3K4me^3^ at the miR-21 promoter [98]. Physiological miR levels derepressed the expression of PDCD4, which inhibited proliferation and increased cell differentiation. Upon ectopically restoring KDM5A expression in CML cells, differentiation was restored, and proliferation inhibited. This study suggests the potential use of KDM5A level as biological marker for CML progression towards blast phase. Further investigation will be necessary to provide evidence for a tumor suppressive function for KDM5A in CML.

Paralleling reports from solid cancers, Garcia and colleagues found that the chemo-resistant AML cell lines Molm13 and Jurkat could be re-sensitized to AZD1775 treatment [114,115] by knockdown of KDM5A, resulting in cell death. This was achieved either directly by using the KDM5 inhibitor CPI-455 or indirectly by inhibition of HDACs which negatively regulate the activity of KDM5A [108].

Together, these findings suggest an important role for KDM5A in disease progression and drug resistance (Figure 2). Future investigation and more selective drug development will be necessary to determine clinical applicability of KDM5A inhibitors especially in accelerated CML, for which few therapeutic options exist.

#### 2.4.2. KDM5B

Similar to KDM5A, evidence suggests a pro-leukemic role for KDM5B in myeloid malignancies (Figure 2). KDM5B is required for HSC self-renewal in mice [49,50] and is expressed in human CD34^+^ cells and in the CML cell line K562, as well as in several AML cell lines (Kasumi-1, KG-1, HEL, HL60, MonoMac-60) (Table 1) [46]. Knockdown of KDM5B in K562 cells has shown to reduce colony-forming potential [46]. Recent expression data from CML patients show that during chronic phase KDM5B level are not changed compared to healthy controls, but significantly increase in CML-BP [116]. Follow-up studies in mouse models of CML-BP will be necessary to determine the functional relevance of this overexpression.

In AML, Wong et al. demonstrated that KDM5B negatively regulates leukemogenesis in both mouse and human MLL-rearranged AML cells [47] through H3K4 demethylation, which leads to cell differentiation. Orgueira and colleagues used a computer learning algorithm termed ST-123 to predict survival of AML patients [48]. Their results suggest that, aside from age, expression of KDM5B and LAPTM4B are the strongest predictors of overall survival using multivariate cox regression analyses. The authors argue that KDM5B may regulate the expression of several oncogenes and tumor suppressors, witnessed in both solid tumors, as well as in leukemia [117,118,119]. Together, these findings highlight a role of KDM5B in myeloid malignancies as predictive marker and a therapeutic target. A first approach to exploring a therapeutic window could consist of determining a potential selective vulnerability of KDM5B-overexpressing leukemic cells versus healthy CD34^+^ HSCs. This is crucial as KDM5 expression is required for normal stem cell function and therapeutic applicability might thus be hampered by toxicity.

### 2.5. The KDM6 Family

The KDM6 subfamily of the Jumonji domain-containing histone demethylases encompasses three members, KDM6A, KDM6B, and KDM6C. As reviewed by Markolovic et al., all enzymes possess catalytic activity towards mono-, di-, and tri-methylated lysine residues [8]. H3K27 is the known main histone target of KDM6 demethylases [8]. KDM6A contributes to myeloid malignancies in an enzymatic (H3K27me^3^ demethylation) [120] and in a non-enzymatic [55] manner. KDM6B demethylates H3K27me^3^ and increases H3K4me3 levels as it associates with the H3K4 methyltransferase complex [60,61,65]. No major contribution of KDM6C to myeloid malignancies has been described, which is why we focus on KDM6A and KDM6B in the following section.

#### 2.5.1. KDM6A

In myeloid malignancies KDM6A is frequently mutated, resulting in reduced expression or loss of function [121,122,123,124,125]. KDM6A deficient mice display a CMML/MDS-like phenotype that frequently transforms into AML (Table 1) [53,54,55]. Mechanistically, KDM6A is required for differentiation [120]. In cord blood CD34^+^ HSCs, KDM6A maintains low H3K27me^3^ levels, allowing binding of the lineage-determining transcription factors CEBPA, SPI1/PU.1 and GATA1 [120]. Increased H3K27me^3^ through KDM6A inhibition prevents recruitment of these transcription factors and results in impaired differentiation [120].

Gozdecka et al. revealed that KDM6A also exerts its tumor suppressive role through noncatalytic mechanisms [55]. KDM6A interacts with members of the COMPASS complex. Loss of KDM6A results in malfunctioning of this complex, causing reduced H3K4 methylation leading to upregulation of oncogenic ETS and repression of tumor suppressive GATA transcriptional programs [55]. Importantly, ETS factors are known to recruit histone acetyltransferases and consequently increase H3K27ac deposition [55,126]. Work by Stief et al. revealed that H3K27ac accumulation upon loss of KDM6A confers cytarabine resistance in AML through downregulation of the nucleoside membrane transporter ENT1 [56]. Moreover, Wu et al. showed that the KDM1A/LSD1 inhibitor SP2509 reverses the differentiation block mediated by KDM6A deficiency. These authors propose a model in which both LSD1 and the malfunctioning COMPASS-like complex demethylate H3K4. LSD1 inhibition increases H3K4 methylation and restores differentiation [127]. In summary, loss of KDM6A contributes to leukemogenesis and drug resistance in myeloid malignancies both dependent on and independent of its enzymatic activity (Figure 2).

KDM6A may also possess oncogenic potential in CML and AML (Figure 2), as high expression was observed in patients with poor prognosis [57,59,128]. In CML, KDM6A promotes imatinib-resistance through upregulation of TRKA, a high affinity receptor for the growth factor NGF [57]. Pro-survival TRKA/NGF signaling is associated with therapy resistance in hematological disorders [129,130,131,132,133]. Targeting KDM6A in CML cell lines and primary cells reduced TRKA levels and sensitized cells towards imatinib-induced apoptosis [57]. KDM6A is also required for cell expansion as its depletion significantly reduced proliferation in AML cell lines [58]. Mechanistically, KDM6A diminishes H3K27 promoter methylation of target genes, resulting in increased expression [58,59]. Targets include Runx1, Mll1, and Scl, frequently deregulated in AML [58]. Moreover, KDM6A elevates levels of the guanine exchange factors DOCK5/8 that convert Rac-GDP to Rac-GTP [59]. Signaling mediated by Rac GTPases promotes cancer cell survival in AML [134,135,136]. To conclude, these studies demonstrate an oncogenic role for KDM6A and provide a rationale for targeted therapies.

#### 2.5.2. KDM6B

In unbiased whole-genome wide ChIP-seq analyses of primary MDS CD34^+^ BM samples, Wei et al. were able to identify 36 genes with elevated H3K4me^3^ levels at their promoters (Table 1) [60]. KDM6B was overexpressed in these cells [60]. Functionally, shRNA-mediated knockdown of KDM6B in primary MDS samples increased the numbers of erythroid colonies [60]. In more recent studies, overexpression of KDM6B in transgenic mice let to a mild hematopoietic phenotype with features of MDS and CMML, including dysplasia, leukopenia, and an impaired repopulation capacity of HSPCs [62]. KDM6B is also overexpressed in AML where its levels correlated positively with poor survival [61]. GSK-J4, a KDM6B inhibitor, showed efficacy in both the AML cell line Kasumi-1, as well as in primary AML patient cells, by decreasing proliferation and colony formation, respectively [61]. Moreover, GSK-J4 treatment of a human AML xenograft mouse model halted disease progression. Mechanistically, treatment enriched H3K27me^3^ at HOX gene promoters, abrogating their expression [61]. Fittingly, Mallaney et al. reported that a complete loss of Kdm6b severely impaired HSCs stem cell self-renewal and prolonged survival of MLL-AF9 leukemic mice [63]. Likewise, KDM6B knockdown re-sensitized AML cell lines to daunorubicin and cytarabine by downregulation of IL6 [64]. Together, these studies clearly suggest a potential for KDM6B as a drug target in MDS and AML.

Although the abovementioned studies focus on KMD6B as a potential oncogene in disease initiation and progression of myeloid malignancy, Yu et al. found a significant role of KDM6B in the differentiation of AML FAB subtypes M2 and M3. Combining ChIP and RNA-sequencing the authors show that KDM6B enables the expression of myeloid differentiation genes. Mechanistically KDM6B reduces H4K27me^3^ and increases H3K4me^3^ levels through its demethylase activity and association with the H3K4 methyltransferase complex, respectively [65]. CEBPB is a KDM6B target gene and expression of the two genes is correlated in primary AML blasts [65]. Knockdown of CEBPB reversed the effect of KDM6B on myeloid gene expression, cell-cycle arrest, and cell death [65]. As shown for other KDMs, whether KDM6B functions as an oncogene or as a tumor suppressor is strongly disease stage and cell lineage dependent (Figure 2).

### 2.6. The KDM7 Family

The KDM7 subfamily of the Jumonji domain-containing histone demethylases encompasses three members, KDM7A, KDM7B, and KDM7C. As reviewed by Markolovic et al., all enzymes possess catalytic activity towards mono-, di-, and tri-methylated lysine residues [8]. H3K9, H3K27, and H3K20 are the known main histone target of KDM7 demethylases [8]. The role of KDM7A-C in myeloid malignancies remains unknown.

## 3. Discussion

Recent research on JmjC domain-containing histone demethylases has revealed clear contributions to disease initiation and progression in myeloid malignancies. However, this emerging field faces several challenges including heterogeneity, potential functional redundancy, and substantial context-sensitivity of function both between KDM families and among members of individual families. For example, KDM5B is required for normal HSC function as witnessed by compromised reconstitution of *Kdm5b* knockout PB cells in a BM transplantation [49]. However, when *KDM5B* is constitutively knocked out, it is dispensable for HSC function. This raises the question whether other KDMs compensate for embryogenic loss of this enzyme. Another obstacle in defining the role of individual KDMs derives from contradictory, context-dependent findings. For example, KDM2B has been described as oncogenic by repressing the tumor suppressor p15 in AML [12]. However, other groups have found that low KDM2B expression was associated with poor survival in AML patients [19,70,71]. Likewise, in a mouse model of AML, low KDM2B expression resulted in shortened survival while overexpression significantly extended the survival [19].

Evidence suggesting oncogenic potential has led to the development of pharmacological KDM inhibitors. Future studies will determine if inhibition of a single KDM is sufficient to reduce leukemic growth or whether other KDMs compensate, abrogating any effect. Moreover, is there a therapeutic window, can healthy cells be spared? Given the high degree of similarity between individual KDMs, it seems likely that inhibitors will affect multiple members of the same family. Nonetheless, with careful drug design and optimization, KDM inhibition might serve to overcome non-genetic drug resistance in myeloid malignancies [137,138,139].

## Figures and Tables

**Figure 1 biomolecules-11-01911-f001:**
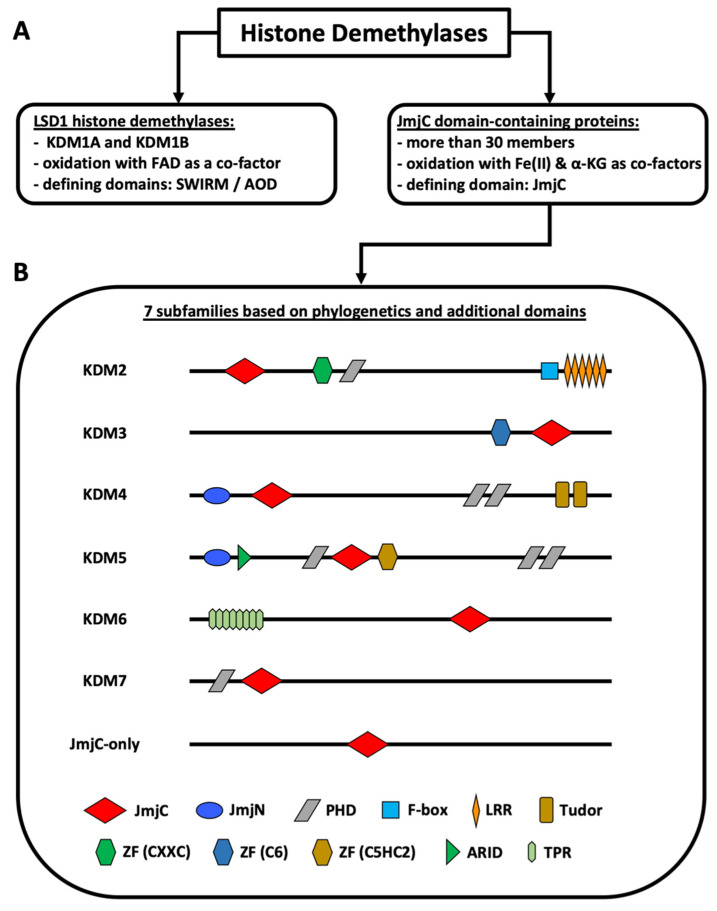
Classification of histone demethylases. (**A**) Types of histone demethylases. (**B**) Subfamilies of JmjC domain-containing proteins. The KDM2–7 subfamilies are known histone demethylases. The JmjC-only subfamily contains demethylases, hydroxylases, and proteins with unknown functions. KDM4D+E lack the PHD and the Tudor domains. KDM5C+D only possess one C-terminal PHD domain. KDM6B lacks the TPR domains. The figure is based on Klose et al. [1], Chen et al. [6], Franci et al. [7], Markolovic et al. [8], and Chang et al. [9]. Abbreviations: αKG (alpha-ketoglutarate), AOD (amino oxidase domain), ARID (AT-rich interaction domain) domain, FAD (flavin adenine dinucleotide), JmjC (Jumonji C) domain, JmjN (Jumonji N) domain, LRR (leucine-rich repeat) domain, PHD (plant homeodomain) domain, TPR (tetratricopeptide repeat domain), ZF (zinc finger) domain.

**Figure 2 biomolecules-11-01911-f002:**
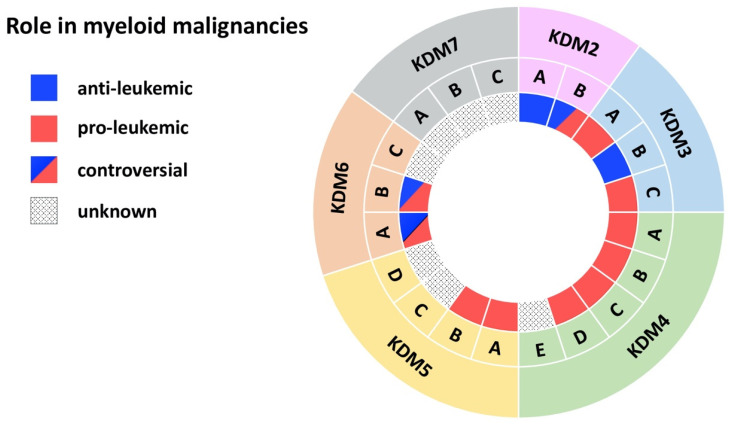
JmjC domain-containing histone demethylases in myeloid malignancies. The outer ring names the KDM subfamily, while the middle ring specifies members of each subfamily. The inner ring indicates the role in myeloid malignancies: blue (anti-leukemic), red (pro-leukemic), blue and red (controversial role), black dots (unknown role).

**Table 1 biomolecules-11-01911-t001:** Involvement of JmjC domain-containing histone demethylases in myeloid malignancies.

Demethylase	Disease	Model	Effect	Mechanism	Reference
KDM2A	AML	MLL-AF10-induced leukemia in mice	anti-leukemic	KDM2A antagonizes oncogenic LEDGF/ASH1L	[10]
	AML	Chemically induced leukemia		KDM2A is downregulated in benzene-induced AML cells	[13]
KDM2B	AML	AML patient cells; MLL-AF9 transduced CD34+ cells; mouse xg models	pro-leukemic	KDM2B as part of the PRC1.1 complex regulates LDHA/PKM independent of H3K27me^3^	[16]
	AML	Tg mouse model with Kdm2b overexpression		KDM2B induces leukemia by increasing expression of Nsg2 and OXPHOS genes	[17]
	AML	AML cell lines; AML CD34^+^ primary cells		KDM2B promotes cell cycle progression by reducing the tumor suppressor p15	[11]
	AML	AML patient cells; *Hoxa9/Meis1-induced leukemia*		KDM2B promotes leukemic transformation by reducing the tumor suppressor p15	[12]
	MDS	Primary MDS cells, MDS cell lines	anti-leukemic	KDM2B suppresses EZH2 through miRNA let-7b expression	[18]
	AML	KrasG12D mice		KDM2B interacts with PRC1/2, increases Irf+Stat, downregulates Hoxa10+Smarca4/Brg1	[19]
KDM3A	AML	Primary AML patient cells	pro-leukemic	KDM3A is recruited by Oct1 to the CDX2 promoter to remove repressive H3K9me2	[20]
KDM3B	APL	NB4 APL cell line	anti-leukemic	KDM3B kd enhances proliferation, blocks differentiation, inhibits degradation of PML/RARα	[21]
	AML	Primary AML patient cells; AML cell lines		KDM3B is downregulated in AML/MDS and overexpression represses colony formation	[22]
	AML	AML cell lines		Expression of KDM2B reduces leukemic growth	[23]
KDM3C	AML	AML cell lines	pro-leukemic	KDM3C modulators selectively inhibit the growth of leukemic stem cells	[24]
	Ph+ MPN	K562 and MEG-01 cell lines		KDM3C kd impairs proliferation, viability, and sensitivity towards chemotherapy	[25]
	Ph- MPN	Jak2V617F mice		Loss of Kdm3c is dispensable for disease initiation	[26]
	AML	AML; MLL cell lines		The Kdm3c inhibitor JDI-16 induces apoptosis and differentiation	[27]
	AML	Mouse MLL-AF9 leukemia cells		Loss of Kdm3c activity increases apoptosis+differentiation via RAS/MAPK, JAK-STAT, IL3	[28]
	AML	HOXA9/MEIS1 bone marrow transplantation model		Kdm3c upregulates key glycolytic and oxidative genes independent of its enzymatic activity	[29]
	Ph-MPN	Primary MPN cells; NFE2 overexpressing mice; MPN cell lines		KDM3C and NFE2 form a positive feedback loop	[30]
	AML	MLL-AF9 and HOXA9 leukemia mice		KDM3C interacts with HOXA9 and supports a HOXA9-controlled gene-expression program	[31]
	AML	AML cell lines		KDM3C is recruited by RUNX1–RUNX1T1 to maintain low H3K9me2 at its target genes	[32]
	AML	MLL-AF9 Tx mouse models; AML cell lines		Depletion of Kdm3c increases apoptosis of leukemic cells	[33]
KDM4A-C	AML	MLL-AF9 mouse model and cell lines	pro-leukemic	Combined KDM4 demethylase activity promotes survival of leukemic cells and increases expression of Il3ra	[34]
KDM4A	AML	Primary AML patient cells; AML cell lines; mouse xg models	pro-leukemic	Loss of KDM4A induces apoptosis and downregulates pro-leukemic gene expression	[35]
	APL	NB4 APL and other cancer cell lines		KDM4A inhibitors increase H3K9/H3K36 methylation and kill malignant cells	[36]
KDM4B	AML	MLL-AF9 transduced CD34+ cells; Primary AML patient cells; AML cell lines	pro-leukemic	KDM4B supports proliferation through upregulation of S100A8/9 and loss of KDM4B reduces growth of leukemic cells	[37]
KDM4C	AML	Primary AML patient cells; AML cell lines; mouse xg models	pro-leukemic	KDM4C regulates miR-328-3p/CCND2 through MALAT1 resulting in Ara-C resistance	[38]
	AML	Primary AML patient cells; AML cell lines; mouse xg models		KDM4C upregulates ALKBH5 resulting in increased AXL mRNA stability	[39]
	AML	Leukemic cells with MLL fusions and MOZ-TIF2; mouse xg models		KDM4C regulates target genes of MLL fusions/MOZ-TIF2 via H3K9me^3^ demethylation	[40]
	AML	AML cell lines		KDM4C mediates oncogenic activity of PRL-3 by reducing H3K9me^3^ at the Leo1 promoter	[41]
KDM4D	AML	AML cell lines	pro-leukemic	KDM4D promotes proliferation in AML cells and activates expression of MCL-1 through H3K9me^3^ demethylation	[42]
KDM5A	AML	Mouse NUP98 fusion-induced leukemia	pro-leukemic	NUP98-KDM5A (PHD finger) fusions induces differentiation arrest and leukemia	[43]
	CML	K562 cells		KDM5A kd in CML-BP stimulates leukemia cell differentiation and inhibits cell proliferation	[44]
	CML	K562 cells, primary patient samples		miR-181d downregulates KDM5A which inhibits NF-κB subunit, p65	[45]
KDM5B	AML/CML	CD34+ cells, AML and CML cell lines	pro-leukemic	KDM5B is highly expressed AML/CML cells, kd reduced leukemia colony-forming abilities	[46]
	AML	Mouse MLL-AF9/10 leukemia cells, MLLr patient samples		KDM5B negatively regulates leukemogenesis	[47]
	AML	Clinical data		KDM5B expression predict survival	[48]
	AML	Mouse		KDM5B is required for hematopoietic stem cell self-renewal	[49,50]
KDM5C	AML	Primary AML patient cells (M5)	unknown	KDM5C is overexpressed in pediatric AML (M5)	[51]
	AML	Primary AML patient cells		KDM5C is mutated and enriched in chemotherapy-resistant pediatric leukemia	[52]
KDM6A	CMML	KDM6A ko mice	anti-leukemic	Loss of KDM6A causes an CMML-like disease	[53]
	MDS	KDM6A ko mice		Loss of KDM6A causes myelodysplasia	[54]
	AML	KDM6A ko mice, AML cell lines		KDM6A ko causes COMPASS complex malfunctioning with upregulation of ETS signaling	[55]
	AML	Primary AML patient cells; AML cell lines		Loss of KDM6A confers cytarabine resistance through ENT1 downregulation	[56]
	CML	Primary CML patient cells; CML cell lines	pro-leukemic	KDM6A promotes imatinib-resistance through upregulation of TRKA	[57]
	AML/CML	AML and CML cell lines		KDM6A depletion reduces *Runx1*, *Mll1* and *Scl* expression and impairs proliferation	[58]
	AML	Primary AML patient cells; AML cell lines		KDM6A promotes cancer cell survival via upregulation of DOCK5/8	[59]
KDM6B	MDS	Primary MDS patient CD34+ cells	pro-leukemic	Inhibition of KDM6B resulted in an increase in erythroid colonies in MDS	[60]
	AML	Clinical data		KDM6B is overexpressed in AML and correlates with a. poor survival	[61]
	MDS/CMML	Tg KDM6B overexpression in mice		KDM6B overexpression showed features of MDS in mice	[62]
	AML	Kdm6b ko (VAVCre, MxCre, ERT2-Cre)		Loss of Kdm6b reduced HSCs and attenuates MLL-AF9-induced AML	[63]
	AML	AML cell lines		KDM6B kd reduced the proliferation and increased chemo-sensitivity	[64]
	AML	HL-60; primary patient samples; PML-RARα-, AML1-ETO9a, or MLL-AF9 tg mice	anti-leukemic	KDM6B exerts anti-AML effect by directly modulating H3K4 and H3K27	[65]

AML (acute myeloid leukemia), APL (acute promyelocytic leukemia), BP (blast phase), CMML (chronic myelomonocytic leukemia), CML (chronic myeloid leukemia), HSC (hematopoietic stem cell), kd (knockdown), ko (knockout), MDS (myelodysplastic syndrome), MPN (myeloproliferative neoplasms), tg (transgenic), xg (xenograft).

## Data Availability

This manuscript reviews published data. Therefore no original data that could be made available was generated.

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
