# Peer review of "The Cross Marks the Spot: The Emerging Role of JmjC Domain-Containing Proteins in Myeloid Malignancies"

_biomolecules, 2021, doi:10.3390/biom11121911_

Round 1

Reviewer 1 Report

Review: biomolecules-1498771

Staehle HF, Pahl HL, Jutzi JS: The cross marks the spot: the emerging role of JmjC Domain-2 Containing proteins in myeloid malignancies

In their manuscript, the authors review the current knowledge about the function of Jumonji C domain-containing histone lysine demethylases in myeloid malignancies. They provide a deep insight into state of the art research regarding these molecules, with a focus on potential therapeutic interventions.

I found the manuscript well written and have only a few comments.

1) For readers not familiar with this family of molecules a simple overview over their molecular structure would be helpful.

2) I missed a paragraph about KDM7 molecules, although they are mentioned in the introduction and included in figure 1.

3) I missed general comments on whether KDMs regulating repressive histone marks have a distinct function compared to those regulating activating histone marks etc.

4) I do understand that figure 1 cannot include all the targets of the demethylases.  Thus, it would be helpful, if the targets are introduced in the header of each subgroup, as e.g. for the KDM2 family (and in the figure explaining the structure)

Minor:
Line 59: Please introduce MLL genes
Line 68: KDM2B is introduced as a H3K36me3 HDM, but described are functions as H3K36me2 HDM please comment on that
Line 105, 188 see 4)
Line 128 is that actually a functional link or just correlation
Line 151 Which methylation state?
Line 190 not ‘one laboratory’ just the ‘Helin lab’ 
Line 199-204 please improve the logic of the passage e.g. ‘in contrast to Helin et al.’
Line 263 Which ‘repressive histone marks’? 
Line 337: consider a line break. You start describing negative regulation.
Line 345 ‘emerging’?
Line 353 also described (line 397) as H3K4 HDM
Line 445-450 Why do they provide examples in the discussion?
Line 455 see 1)

Author Response

The authors thank the reviewer for the fast, positive, and detailed response to our manuscript. Please find our responses attached.

Sincerely,

Jonas Jutzi

Reviewer 2 Report

This is a very comprehensive literature review about histone demethylases. The main issues with respect to that is the layout and the structure of the review rather than the explicit content itself. 

I would suggest a revision and some editing of the English to make it more cohesive and readable. I can highlight some portions to provide some examples. Overall, I have the following suggestions:

  1. Make a more complete intro for the subject and your Figure 1. In the Intro say that JmjCs are not only involved  in tumor suppressor function, but also enhancement. I think that it should be rather stated that there is a diversity in their types and classification and often a conundrum about their exact function. why not mention it here in the intro and explain the two differences in the respective sections? Another thing that could be included in the introduction is the subclassifications of the KDMs in A, B, C, etc. and what does this subclassifications refers to. also there is a significant misnumbering. In the Intro it is referred that the JmjCs comprise KDM2-8, then the authors state that they will described KDM2-7 and the manuscript ends at KDM6.

2. In places that you say that further investigation is needed, Specify or be more clear about why these data are not adequate.

3. there are references to a lot of proteins in the text and someone not relevant to the filed may be overwhelmed. You should create a table of supplement with the abbreviations of these proteins and their functions. 

4. For the better flow of the ms I would suggest the following. The authors in each subsection they describe the general action of the KDMn, where n is a number. Going into the more specific subsections, the description of what is known in the literature is a bit convoluted and does not really seem to build up. I would  suggest a common structure for all modules that is currently missing. Maybe starting with the type of demethylase acivity  of the exact demethylase and then say with what cellular functions 
it has been associated and if that leads to enhancement of the suppression or enhancement of tumorigenesis.

Author Response

(The authors gave the same response as above.)
